# Influence of Caffeic and Caftaric Acid, Fructose, and Storage Temperature on Furan Derivatives in Base Wine

**DOI:** 10.3390/molecules27227891

**Published:** 2022-11-15

**Authors:** Jacob Medeiros, Shufen Xu, Gary J. Pickering, Belinda S. Kemp

**Affiliations:** 1Department of Biological Sciences, Faculty of Mathematics and Science, Brock University, St. Catharines, ON L2S 3A1, Canada; 2Cool Climate Oenology and Viticulture Institute (CCOVI), Brock University, St. Catharines, ON L2S 3A1, Canada; 3National Wine and Grape Industry Center, Charles Sturt University, Wagga Wagga, NSW 2678, Australia; 4Sustainability Research Centre, University of the Sunshine Coast, Sippy Downs, QLD 4556, Australia

**Keywords:** base wine, caffeic acid, caftaric acid, fructose, furan derivatives, temperature

## Abstract

The aim of this study was to determine the influence of caffeic and caftaric acid, fructose, and storage temperature on the formation of furan-derived compounds during storage of base wines. Base wines produced from Chardonnay grapes were stored at 15 and 30 °C for 90 days with additions of fructose, caffeic acid, and caftaric acid independently or in combinations. Wines were analyzed following 90 days of storage for: total hydroxycinnamic acids, degree of browning, caffeic acid and caftaric acid concentrations, and nine furan-derived compounds. Caffeic and caftaric acid additions increased homofuraneol concentration by 31% and 39%, respectively, at 15 °C (*p* < 0.05). Only the addition of caffeic acid increased furfural by 15% at 15 °C (*p* < 0.05). Results demonstrate that some furan derivatives over 90 days at 15 °C increased slightly with 5 mg/L additions of caffeic and caftaric acid. This is the first time the influence of hydroxycinnamic acids on furan-derived compounds has been reported during short-term aging of base wine at cellar temperature.

## 1. Introduction

The first stage of sparkling wine production is the primary alcoholic fermentation to produce a base wine from whole bunch pressed, fractionated grape juice [1]. It is in the second stage of the traditional method of winemaking that the base wine gives rise to sparkling wine [2]. Reserve wines (base wines aged after primary fermentation) are stored under specific conditions and used by Champagne producers to produce non-vintage or multi-vintage sparkling wines [3]. This typically involves blending several base wines from different vintages, varieties, and/or regions to produce the final wine. These reserve wines can be stored for extended periods—one to thirty-five years, and in some cases even longer [3].

During the ageing and storage of “reserve wines”, with or without contact with yeast lees (contact time can differ by winery), in a temperature-controlled environment (12–15 °C), changes in aroma and flavour that distinguish the mature base wine from a more youthful wine occurs [2,3]. Amongst these changes, are a change of flavour to a distinctive empyreumatic character, reminiscent of toasted bread, roasted nuts, and caramel-like aromas [4,5,6]. Le Menn et al. [6] reported compounds responsible for these distinctive characters in reserve base wines. For instance, 5-methylfurfural, responsible for caramel and nutty characteristics in wine, is a well-known product of the Maillard reaction (MR) [5,6].

MR is a non-enzymatic set of chemical reactions categorized by the initial reaction of a reducing sugar with an amino acid, protein and/or peptide, generating a cascade of thousands of compounds [6,7,8]. Research has shown that despite the limiting conditions for the MR to occur in wine [9], products of MR have been found in aged sparkling wines [6,10,11,12,13]. Specifically, a group of compounds known as furans (O-heterocyclic 5-membered aromatic ringed compounds) have been highlighted in sparkling wine ageing [6]. Torrens et al. [14] analyzed Cava wines that had aged for 24 months, using Gas Chromatographic-Olfactometry (GC-O) and reported that furans including furfural, 2-methyl-3-furanthiol, 5-methylfurfural, 2-furanmethanethiol, and 2-acetylfuran contribute “fruity/caramel”, “toasty”, “fruity/caramel”, “dried fruit”, and “balsamic” aromas respectively. All furan-derived compounds identified by Torrens et al. [14] were detected more frequently and with greater intensity in an aged Cava wine compared to a younger base wine. This indicates that their presence is associated with the wine maturation process.

In wine, factors that affect the formation of MR products include temperature, time, pH, wine chemical composition, and sulfur dioxide (SO_2_) levels, but little is known about the role of specific amino acids, metal ions, sugars, and phenolic compounds such as hydroxycinnamic acids (HCAs), especially in base wines [9]. HCAs are characterized by an aromatic ring with a three-carbon side chain (C6–C3) and are the most abundant phenolic compounds in still white wines, specifically caffeic acid and its tartaric acid ester, caftaric acid [15,16]. In sparkling wines, caffeic acid concentrations were found to range between 0.2–2.0 mg/L, while caftaric acid was found within a greater range of 2.2–31 mg/L [17,18,19,20]. Concerning their relationship to the Maillard reaction associated compounds; Zhang et al. [21] studied caffeic acid and fructose in a phosphate-buffer model solution at pH 5.5 that was heated to 90 °C. The authors reported that the presence of 5 µ mol/mL of caffeic acid increased the generation of the MR-associated compounds, specifically 5-hydroxymethylfurfural (HMF) and furfural, compared to heating the fructose solution alone. However, the study used model wines at a higher pH level than in wine and at a higher temperature than that used in winemaking.

Artificially accelerating the ageing of wines, such as in the aforementioned study by Zhang et al. [21], is mainly conducted by heating the wines to hasten the rate of chemical reactions [22,23]. This allows for determination of changes in age-related aroma compounds in a more timely fashion [19,23,24]. However, this approach has its limitations, as some compounds associated with MR, such as 5-hydroxymethylfurfural (HMF), are formed more rapidly at higher temperatures compared to typical cellar temperature (12–16 °C) [25]. Reineccius [26] identified that the formation of MR associated products (and their respective aromas) would not be identical for different storage times and temperature settings [9]. This highlights the need for research on the formation of MR-associated products (e.g., furan-derived compounds) in wine in representative wine-like temperature conditions. Additionally, the degree to which HCAs, fructose, and storage temperature contribute to the formation of MR-associated products in base wines requires investigation due to the lack of knowledge concerning this part of wine ageing.

The aim of this study was to determine the effect caffeic and caftaric acid, fructose, and temperature have in base wine on the synthesis of nine furan-derived compounds previously identified in sparkling wines.

## 2. Results

### 2.1. Caffeic and Caftaric Acids

An interaction between treatment and temperature on caffeic acid concentration was revealed (*F* (DF 15) = 4351, *p* < 0.0001). Between temperature conditions, samples that did not have caffeic acid added, and were kept at 15 °C had lower concentrations of caffeic acid after 90-days compared to the 30 °C samples, although their initial concentrations were the same (0.23 mg/L, *p* < 0.05). The caffeic acid addition treatments (CAFE and F + CAFE) stored at 15 °C retained a higher concentration of the acid compared to those kept at 30 °C (0.6 mg/L, *p* < 0.05). This relative decrease in caffeic acid concentration in the 30 °C samples (10–15%) was only observed in treatments where caffeic acid was added independent of caftaric acid (Figure 1).

Caftaric acid concentrations were impacted by treatment and storage temperature (*K* (DF 15) = 56.5, *p* < 0.001). Concentrations of caftaric acid in those treatments without 5 mg/L caftaric acid addition (CNTL, FRU, CAFE, and F + CAFE) were found to be at 20.8, 21.3, 20.7, and 20.3 mg/L respectively. Those treated with caftaric acid (CAFT, F + CAFT, CAFE + CAFT, and F + CAFE + CAFT) are reported at 25.9, 26.1, 27.5, and 27.8 mg/L respectively (Table 1). Between 0 and 90 days, a trend of increasing caftaric acid was found; the overall average concentration of caftaric acid in unmodified samples increased from 20.9 mg/L to 26.3 mg/L (Table 1). This trend was observed for caftaric acid-addition samples as well, with an initial average of 27.0 mg/L compared to the 90-day sample average of 33.4 mg/L. Results for total hydroxycinnamic acids (HCAs) and the estimated degree of browning can be found in Appendix A.

Between temperature conditions, caftaric acid concentration increased to a greater extent at the 15 °C condition compared to the 30 °C condition, with all 15 °C treatments exhibiting a higher caftaric acid content except for F + CAFE + CAFT, which was the same concentration of 33–34 mg/L (*p* < 0.05).

### 2.2. Furan-Derived Compounds

Nine furan derivatives that had previously been detected in base wines, sparkling wines, and still wine styles, were analyzed using HS-SPME-GC-MS. Only three of the compounds were present in all samples at concentrations greater than the limit of quantification (LOQ) of the method; furfural, ethyl-2-furoate, and homofuraneol (Table 2).

#### 2.2.1. Furfural

Significant variation in furfural concentration was revealed based on the treatment applied for the 30 °C samples (*F* (DF 7) = 29.5, *p* = 0 < 0.0001) as well as the 15 °C samples (*F* (DF 7) = 6.3, *p* < 0.014). An increase of approximately 80 µg/L in all samples in the 30 °C condition was observed compared to those kept at 15 °C corresponding to a 164% increase in furfural concentration (Figure 2). Within the 30 °C sample set, furfural concentration was significantly greater in all treatment conditions compared to the control (7 µg/L, *p* < 0.05) (Table 2). For the 15 °C samples, a less uniform change was observed; the addition of 5 mg/L of caffeic acid resulted in a 15% increase in furfural concentration compared to the control (*p* < 0.05). No difference was observed for the other treatments (*p* > 0.05) (Table 2).

#### 2.2.2. Ethyl-2-Furoate

Treatments did not have a significant effect on ethyl-2-furoate concentrations at 15 °C (*F* (DF 7) = 2.3, *p* = 0.057), though differences in the 30 °C samples were significant (*K* (DF 7) = 21.0, *p* < 0.005). A greater concentration of ethyl-2-furoate was observed at the 15 °C storage temperature (2.6–2.9 µg/L) compared to the storage condition at 30 °C (1.5–2.4 µg/L). This difference was greatest in the control, which had 52% less ethyl-2-furoate in the 30 °C wines compared to those stored at 15 °C (Figure 2). Within temperature conditions, the 15 °C wines did not vary across treatments under the statistical model applied (*p* > 0.05). However, in the 30 °C storage condition, differences were observed between treatments: caffeic (CAFE) and caftaric (CAFT) acid additions increased ethyl-2-furoate levels by 33% and 36% respectively, relative to the control, and the addition of both compounds (CAFE + CAFT) resulted in a greater increase of 55% relative to the control (*p* < 0.05) (Table 2).

#### 2.2.3. Homofuraneol

For the wines stored at 15 °C, significant differences between treatments (*F* (DF 7) = 12.3, *p* < 0.0001) were found, while no significant differences were reported for the 30 °C samples (*F* (DF 7) = 1.7, *p* = 0.168). Similar to the results obtained for furfural, homofuraneol variation between storage temperatures was observed: the 30 °C condition on average generated 125 µg/L of the furan compound compared to 19 µg/L in the 15 °C condition. This corresponds to a 147% increase between the samples stored at 30 °C and those at 15 °C (Figure 2). Within temperature conditions, the 30 °C sample set showed no variation between treatments at α = 0.05 (Table 2). The fructose addition treatment was not significantly different from any of the non-sugar added treatments. Under the 15 °C storage condition, variation across treatments was found: all treatments increased in homofuraneol levels relative to the control wine at *p* < 0.05 (Table 2). The difference between the 5 mg/L caftaric acid addition (CAFT) treatment and the combined 5 mg/L caffeic and caftaric acid treatment with 6 g/L added fructose (F + CAFE + CAFT) was significant. While the CAFT treatment increased in homofuraneol concentration by 39% relative to the control, the combined F + CAFE + CAFT treatment only increased the homofuraneol concentration by 20%.

#### 2.2.4. 5-Methylfurfural

Only samples stored at 30 °C contained quantifiable concentrations of 5-methylfurfural after 90 days of storage. The CNTL, FRU, and FRU + CAFE + CAFT wines contained concentrations below the LOQ of the method, while the rest of the wines were found at 4.9 or 5.0 µg/L of 5-methylfurfural (Table 2). No influence on concentration was determined by treatment effect (*p* > 0.05).

## 3. Discussion

The aim of this study was to determine the influence of caffeic acid, and its tartaric acid-ester caftaric acid, on the formation of Maillard-Reaction (MR) associated furan-derived compounds during the storage of base wine. To assess this, quantification of the furan-derived compounds was carried out using HS-SPME-GC-MS, in addition to caffeic and caftaric acids, which were determined by LC-UV/DAD.

### 3.1. Caffeic and Caftaric Acids

The decrease in the concentration of caffeic acid, alongside the reported increase in caftaric acid in all wine samples regardless of temperature, is in contradiction to previously published results that assessed HCA concentrations over time [17,27]. In both cited studies, the caftaric acid concentration decreased over the course of the experimental timeline. This could be due to the oxidation of caftaric acid into its *o*-quinone form, or through combination with glutathione to form 2-*S*-glutathionyl caftaric acid (Grape Reaction Product—GRP) [28]. Similar to the behaviour observed for caftaric acid in our study, Serra-Cayuela et al. [17] reported up to a 77% increase in caffeic acid concentration throughout their ageing experiments conducted at multiple temperatures (4, 16, and 20 °C). This observation was explained by Ferreira-Lima et al. [27] as the hydrolysis of the esterified forms of HCAs into the free cinnamic acid.

### 3.2. Furfural

The accumulation of furfural in aged sparkling wines has been reported in previous studies [4,11], as well as in other wine styles [29]. Furfural has been described as contributing “sweet”, “bready”, and “almond” aromas, though its odor detection threshold has only been determined by Ferreira et al. [30] in a model wine solution at 14 mg/L. The formation of furfural via MR has also been outlined in detail in the literature [31], and it is known to form preferentially from fructose under acidic conditions.

Wines stored for 90 days at 30 °C produced concentrations of furfural in the 100 µg/L range for all treatments, and treatments were determined to be greater in furfural content compared to the control. The addition of 6 g/L of fructose did not provide a change in furfural concentration, contrary to its anticipated increase [29]. The addition of 6 g/L of fructose (FRU) in our study resulted in the equivalent production of furfural (98 µg/L) as the addition of 5 mg/L of either HCA added independently (CAFE and CAFT, 97 and 98 µg/L, respectively). That fructose is a precursor compound to furfural has already been established in the literature, though other sugars such as xylose or arabinose (pentoses) are also known contributors to furfural formation via the same pathways [32]. Wang et al. [33] reported the molar conversion of these sugars into furfural as low yielding (<3% mol conversion), even under thermally optimal conditions (water, 150 °C, 50 min). The conversion of glucose into furfural was even more challenging, with a molar conversion of <1% under the same conditions [33]. This is due to the C-C bond cleavage of the 6-carbon glucose molecule required to form the 5-carbon structure of furfural, which is typically performed via a retro-aldol mechanism under acidic conditions [33]. The same chemical mechanism applies to fructose as that of glucose, which could account for the limited impact on furfural generation observed in the 6 g/L fructose addition treatments. Taking into consideration the 8 µg/L greater concentration of furfural observed in the 6 g/L fructose treatment (98 µg/L) compared to the control (90 µg/L), our study reports a molar conversion of only 0.00025% of fructose into furfural under the conditions applied. Considering the limited increase in furfural observed in all treatments with added fructose, and the relatively small change in fructose concentration observed over the 90-day storage periods (<0.5 g/L for both temperature conditions for each treatment), this conversion can be considered credible.

Given that, the initial concentration of fructose in the base wine was 2.2 g/L, with a molar conversion as low as 0.00025%, it can be concluded that the majority of the furfural generated during the 90-day storage period did not come from this precursor. Therefore, the major contributor to furfural formation cannot be determined by this study.

While both HCAs added independently were found to increase furfural generation significantly when compared to the control, the combination of both in the same sample did not have a doubling effect on furfural synthesis. This result could be due to a number of factors regarding the catalytic capability of caffeic/caftaric acid to assist in the formation of furfural: the molecular form the HCA must occupy to function as a catalyst, the reaction step during which it operates, and the precursor compound it acts on [34]. Labauze et al. [34] and Zhang et al. [21] suggested that the diphenol structure and H^+^ donation capability associated with caffeic/caftaric acid could be responsible for this catalytic ability, though no mention of the chemical mechanism was made.

Our results are in agreement with Zhang et al. [21], who reported that caffeic acid significantly increased furfural formation (1.5 mg/L) when a fructose-phosphate buffer solution at pH 5.5 was exposed to overheating conditions (90 °C). Though it has been shown in studies that used aqueous model solutions with polyphenols in it such as epicatechin that the formation of MR-associated products by binding to intermediary compounds responsible for their formation occurs. However, this phenomenon was not observed for caffeic or caftaric acid with regards to furfural formation [35,36]. This is the first study to report the direct relationship between caffeic/caftaric acid and the formation of furan-derived compounds in base wine, albeit by a small amount.

Despite the small increase in furfural concentration observed, its presence in wine has larger implications on empyreumatic aroma generation. The odour detection threshold (ODT) for furfural has been reported at a vastly greater concentration than that found in wine (3 mg/L [37] and 14 mg/L [30]). Therefore, the increased levels observed in the 30 °C storage condition can be considered to have little or no impact on wine sensory perception. However, furfural is known to be a precursor to 2-furfurylthiol, a much more potent furan-derived aroma compound with an ODT of 0.4 ng/L [13]. To what degree the increased 8 µg/L of furfural could contribute to 2-furfurylthiol formation, or other reaction products, can only be speculated here as 2-furfurylthiol was not detected in any of the wines analyzed.

### 3.3. Ethyl-2-Furoate

Considered to be a MR intermediary compound, ethyl-2-furoate is the ethanol-esterified derivative of furfural or furoic acid, the oxidation product of furfural or furfuryl alcohol [32]. Discovered relatively recently in wine, ethyl-2-furoate is not well described in the literature [38], though it has been associated with “vanilla” and “burnt” aroma characteristics in wine [24]. No ODT could be found in the literature for this compound to compare it to the concentrations reported in our study.

Ethyl-2-furoate concentrations were no greater than 3 µg/L in all samples, though significant variation in analyte amounts were detected. Our results lend evidence to the reported formation of ethyl-2-furoate from furfural via the furoic acid intermediate, as the addition of fructose and each HCA increased the furfural content in a similar manner [32]. Therefore, concentrations of ethyl-2-furoate could be considered to be primarily dependent upon furfural synthesis. Under the 15 °C condition, no difference in concentrations was found between treatments at α = 0.05 and consequently no treatment interactions could be reported.

The formation of ethyl-2-furoate was greater under the 15 °C storage temperature compared to 30 °C. The further reaction of ethyl-2-furoate, once formed in the wine media, could have occurred at a more rapid rate at 30 °C compared to 15 °C, leading to the higher concentration obtained in the cooler wines [32]. It is important to note that the differences observed in the ethyl-2-furoate concentrations are <1.0 µg/L across all treatment conditions, and therefore the significance of the increased analyte levels expressed should not be overstated. More research into this compound’s formation and its further reaction schemes are necessary to determine its relative importance in the understanding of MR activity in wine during ageing.

### 3.4. Homofuraneol

A more complex furan-derivative, homofuraneol (2-ethyl-4-hydroxy-5-methyl-3(2H)-furanone), also referred to as ethyl furoate, has been reported in several studies in food and flavour chemistry as part of the group of furanones [13,39]. Found in fruits, furanones are of key interest due to their pleasant aroma contributions at relatively low (µg/L) concentrations [5]. Homofuraneol has been described as smelling like “caramel” and “cotton candy” [13], with a relatively low ODT of 10 µg/L (determined by Kotseridis and Baumes [40] in a model wine solution). Homofuraneol formed at detectable levels in all wines, but was found at greater concentrations in wines stored at 15 °C that had 6 g/L of fructose added. Similar to the results obtained for furfural at 15 °C, homofuraneol concentrations were increased when fructose was added, as well as caffeic and caftaric acid, with the largest increase of 39% (corresponding to approximately 5 µg/L) found in the CAFT treatment.

In a study focused on furaneol during wine ageing, Jarauta et al. [41] showed that its concentration increased from approximately 60 µg/L to 140 µg/L during the first 6 months of ageing in French oak barrels kept at cellar temperature. This range is greater than the concentrations reported our study (15–20 µg/L), though this discrepancy can be explained by the presence of oak barrel ageing in the study by Jarauta et al. [41], which contributes to a greater concentration of furaneol in wine.

### 3.5. 5-Methylfurfural

5-Methylfurfural has been reported frequently in wine studies as a compound correlated with wine ageing and contributing similar empyreumatic aroma characteristics to that of furfural; “caramel”, “burnt sugar”, and “almond” [6,11,24]. Known to be a sugar degradation product similar to furfural, 5-methylfurfural was also found in wines aged in oak barrels [41]. However, oak was not used in our study, so the formation of 5-methylfurfural can only be attributed to sugar degradation/MR schemes.

Our results are in agreement with those reported by Pereira et al. [24], who determined that 4 µg/L of 5-methylfurfural was generated in a dry Tinta Negra wine over three months at 45 °C, while 50 µg/L were generated when the same wine was kept at 70 °C for one month. This supports the understanding that increased temperature confers greater formation of MR-associated products such as 5-methylfurfural [9].

### 3.6. Limitations of the Study

The most impactful limitation of the present study was the time duration available for the ageing of base wine, given that the longer a wine is stored, the greater the opportunity for empyreumatic aroma compound formation as seen in aged sparkling wines [10]. As the timeline for the experiment was monthly as opposed to yearly, the generation of aroma compounds at detectable levels was suboptimal, yet successful for select compounds. In addition to this, the ambient concentration of furan-derived compounds of interest was quite low as a result of the relative youth of the base wine studied, despite the fact that the base wine had been aged for 18 months prior to use. An older, longer aged wine could have served as a more substantial base for the experiment as it would have already contained a detectable amount of the furan-derived compounds, which would have allowed for determination of the initial concentrations. This could have aided in the quantification of the low concentration (µg/L) of compounds, as well as allowed for the determination of concentration changes over time during storage.

Additionally, furan-derived compound concentrations were only determined at the end of the 90-day storage period as opposed to at regular periods during storage, which, if the compounds were above the LOQ during this period, could have allowed for linear regression analysis and further extrapolation of potential future concentrations. Repeating this experiment with an older base or sparkling wine, or ageing the wine for a longer duration at 30 °C, would have provided more data for interpretation.

In addition to the temporal component, several wine chemical factors such as pH, SO_2_, acidity, sugars and amino acids were not varied in this study. These components and their influence on the synthesis of furan-derived compounds during the ageing process were not evaluated. These limitations highlight areas for future research using both base wine and sparkling wine.

## 4. Materials and Methods

### 4.1. Chemicals and Standards

The deuterated internal standards of caffeic acid-d_6_ (CAS 2708298-33-1, ≥99%) and furfural-d_4_ (CAS 1219803-80-1, ≥98%) were purchased from CDN Isotopes (Pointe-Claire, QC, Canada). Furfural ethyl ether (CAS 6270-56-0, ≥97%) was purchased from Fisher Scientific (Hampton, NH, USA). The following compounds were purchased from Sigma-Aldrich Inc. (Oakville, ON, Canada); caffeic acid (CAS 331-39-5, ≥98%), caftaric acid (CAS 67879-58-7, ≥97%), 3-acetyl-2,5-dimethylfuran (CAS 10599-70-9, ≥98%), 2-furanmethanethiol (CAS 98-02-2, ≥98%), 2,3-dihydrobenzofuran (CAS 496-16-2, ≥99%), 2-acetylfuran (CAS 1192-62-7, ≥99%), 5-methylfurfural (CAS 620-02-0, ≥98%), homofuraneol (CAS 27538-09-06, ≥96%), furfural (CAS 98-01-1, ≥99%), sodium chloride (NaCl, ≥99%) and ethyl-2-furoate (CAS 614-99-3, ≥99%). All solvents used were HPLC grade purity. Potassium metabisulphite (KMS) and potassium bitartrate (cream of tartar) were purchased from Scott Laboratories Ltd. (Niagara-on-the-Lake, ON, Canada). D-fructose (≥99%) was purchased from BioShop^®^ Canada Inc. (Burlington, ON, Canada). Milli-Q water was obtained Millipore (Saint-Quentin-en-Yvelines, France).

### 4.2. Winemaking

Chardonnay grapes were hand-harvested from Trius Winery on 19 September 2019 in Niagara-on-the-Lake, Ontario, and whole bunch pressed on site. The juice was then transported to the Cool Climate Oenology and Viticulture Institute (CCOVI) at Brock University for processing according to previously documented winemaking practices [42].

### 4.3. Experimental Design

Fructose, caffeic and caftaric acid were added to Chardonnay wines before storing them for 90 days at 15 °C (cellar temperature) and 30 °C (moderate accelerated ageing). All treatments were duplicated, and each sample underwent duplicate analysis for each chemical parameter. Studies show that caffeic acid and caftaric acid concentrations vary widely in base and sparkling wines: 0.2–2.0 mg/L for caffeic acid and 2.2–31 mg/L for caftaric acid [17,18,19,20]. Consequently, equal additions of 5 mg/L of caffeic acid and 5 mg/L of caftaric acid were chosen to provide a consistent concentration to measure the effect each acid has when its concentration is elevated. A treatment containing an additional 6 g/L of fructose was also included to encourage the formation of furan-derived compounds, as many of these compounds are formed from fructose degradation [32].

The 6 g/L concentration of fructose was chosen to supplement the small residual concentration of fructose present in the base wine (<3 g/L). Fructose is typically the most abundant sugar in wine post-alcoholic fermentation (unless a fructophillic yeast was used for fermentation), and its prolonged presence in base and sparkling wines being aged for extended periods allows for the synthesis of furan-derived compounds. The seven treatments and control were made up as follows: CNTL (no addition), FRU (6 g/L fructose), CAFE (5 mg/L caffeic acid), CAFT (5 mg/L caftaric acid), F + CAFE (6 g/L fructose + 5 mg/L caffeic acid), F + CAFT (6 g/L fructose + 5 mg/L caftaric acid), CAFE + CAFT (5 mg/L caffeic acid + 5 mg/L caftaric acid), and F + CAFE + CAFT (6 g/L fructose + 5 mg/L caffeic acid + 5 mg/L caftaric acid). Replicates were stored in either 15 or 30 °C temperature-controlled rooms for 90 days. Following 90-day storage, the following parameters were analyzed; total HCA estimation (A.U at λ_320nm_), degree of browning (A.U at λ_420nm_), caffeic and caftaric acid (mg/L), and nine furan-derived compounds (µg/L) (furfural, ethyl-2-furoate, homofuraneol, 5-methyl furfural, furfuryl ethyl ether, 3-acetyl-2,3-dimethyl furan, 2-furyl methyl ketone, 2-furfuryl thiol, 2,3-dihydro, benzofuran).

#### Standard Wine Chemical Analyses

Titratable acidity (g/L tartaric acid eq.) and pH were determined using an auto-titrator (Hanna Instruments^®^ HI 84502 Woonsocket, RI, USA). Free and total SO_2_ levels was analyzed using the aspiration method [43]. Ethanol (% *w*/*v*) was measured according to Nurgel et al. [44] using Gas Chromatography (Agilent 6890 model, Agilent Technologies Inc., Santa Clara, CA, USA) coupled to a Flame Ionization Detector (GC-FID). Acetic acid (g/L), L-malic acid (g/L), yeast assimilable nitrogen (YAN mg N/L), and residual sugar levels (mg/L D-glucose & D-fructose) were measured using assay kits: K-ACET 02/17; L-LMALL 06/07; K-PANOPA 08/14; K-AMIAR 12/12, and K-FRUGL 05/17 respectively (Megazyme International Ltd., Wicklow, Ireland). The total HCA estimation and degree of browning was determined according to Iland et al. [43]. The chemical analyses of the base wine were carried out prior to additions of the target compounds (Table 3).

### 4.4. Caffeic and Caftaric Acid Determination by Liquid Chromatography-UV Diode Array Detection (LC-UV/DAD)

#### 4.4.1. Sample Preparation

Wine samples were prepared up to 1 mL in 2 mL amber glass vials (Part 5182-0716, Agilent Technologies Inc., Santa Clara, CA, USA), closed with compatible screw-top caps (Part 5185-5823, Agilent Technologies Inc., Santa Clara, CA, USA). 0.5 mL of wine was combined with 0.45 mL of Milli-Q H_2_O, and 0.05 mL of a 0.1 g/L caffeic acid-d_6_ (CAS 2708298-33-1, ≥99%) internal standard solution.

#### 4.4.2. LC-UV/DAD Method

The LC-UV/DAD method for caffeic and caftaric acid determination was modified from Berry and Henderson [45]—the flow rate was modified from 0.3 mL/min to 0.2 mL/min to reduce pressure. The separation of HCAs, obtained by direct injection of wine, was performed using an Agilent 1260 Infinity LC series system (Agilent Technologies Inc., Santa Clara, CA, USA) equipped with a 66 × 2 mL vial auto sampler (G7129 type) and a Zorbax RRHD SB-C18 analytical column (2.1 mm × 150 mm, 1.8 µm particles). Column temperature was kept at a constant 30 °C. The mobile phase for LC-UV/DAD analysis was a mixture of (A) water with 0.1% formic acid (*v*/*v*), and (B) acetonitrile with 0.1% formic acid (*v*/*v*), flowing at 0.2 mL/min under gradient conditions: starting at 0% B into A (0–3.5 min), increasing to 5% B into A (3.5–7.1 min), increasing again to 15% B into A (7.1–25 min), then increasing further to 40% B into A (25–27 min), finally ramping up to 100% B into A (27–30 min), followed by a decrease back to 0% B into A (30–45 min). The total run time was 45 min with a post-run downtime of 5 min. For the quantification, wine samples without any prior treatment were injected directly (1.5 μL) into the system. Each sample was injected in duplicate. Both compounds were quantified using an external calibration curve obtained from the corresponding standard measured at 280 nm and 325 nm wavelengths.

#### 4.4.3. Analytical Performance of the LC-UV/DAD Method

##### Linearity and Limits of Detection (LOD) and Quantification (LOQ)

The linearity of the LC-UV/DAD method was evaluated using the appropriate concentration range for sparkling wine samples as determined by Serra-Cayuela et al. [17]. An aqueous solution containing 10% (*v*/*v*) of ethanol was spiked with 5 mg/L of internal standard alongside 6 concentration levels of each HCA (excluding 0 mg/L), with samples prepared in duplicate. For caffeic acid the standard concentrations were the following: 0 mg/L, 0.25 mg/L, 0.5 mg/L, 0.75 mg/L, 1 mg/L, 1.25 mg/L, and 1.5 mg/L. For caftaric acid, the concentrations were 10× greater: 0 mg/L, 2.5 mg/L, 5 mg/L, 7.5 mg/L, 10 mg/L, 12.5 mg/L, and 15 mg/L. The calibration curves were generated by plotting the relative peak area (relative response factor (RF)) of the analyte (analyte peak area over internal standard peak area) as a function of the analyte concentration. The linearity for all standards was deemed satisfactory, with regression coefficients (*R*^2^) of 0.98 for caffeic acid and 0.99 for caftaric acid in all cases. Limits of detection (LOD) and quantification (LOQ) were determined via the signal: noise method outlined by [46]: the mean of blank sample replicates was added to the standard deviation of blank sample replicates multiplied by 1.645 (LoB = mean blank + 1.645 (SD_blank_)). The LOD was subsequently calculated by adding the LoB to the value of the lowest concentration analyte replicate standard deviation multiplied by 1.645 (LOD = LoB + 1.645 (SD_low conc. sample_)). The LOQ was estimated as the LOD multiplied by 3.3 to ensure error associated with imprecision of the analytical instrumentation was accounted for. The LODs for caffeic and caftaric acid were 0.07 and 0.46 mg/L, respectively, while the LOQs were 0.17 and 1.0 mg/L, respectively.

##### Repeatability, Accuracy and Specificity

The repeatability of the method was evaluated by comparing the relative standard deviations of the samples tested. The coefficients of variation for all samples were below 10%, which confirms good precision of the method. The accuracy of the method was determined by calculating the percent recoveries for the spiked samples compared to the un-spiked sample. The analyte standards were spiked with either 0.5 or 1 mg/L of caffeic acid, and either 5 or 10 mg/L of caftaric acid in a real wine sample. These spiked samples were then compared to the corresponding real wine sample, which had not been spiked, and the relative concentration of the analyte in the spiked sample was compared to that of the un-spiked sample. Recovery for caftaric acid (106.0%) was considered appropriate. The relatively similar concentration of the spike (0.5 mg/L) to that of the endogenous caffeic acid concentrations is thought to have influenced the recovery percentage for the caffeic acid (137.1%) samples, given that the amount of caffeic acid present was already quite low.

### 4.5. Headspace Solid Phase Micro-Extraction-Gas-Chromatography/Mass Spectrometry (HS-SPME-GC-MS) Method for Furan-Derived Compounds

#### 4.5.1. Sample Preparation

Wine samples were prepared up to 5 mL in 10 mL amber glass vials, 1.5 g of sodium chloride (NaCl, ≥99%) was added, and closed with compatible PTFE/silicone screw-top caps (Agilent Technologies Inc., Santa Clara, CA, USA). 4.9 mL of wine sample was combined with 0.1 mL of a 1.0 mg/L furfural-d4 internal standard solution, for a final concentration of 20 µg/L internal standard.

#### 4.5.2. HS-SPME-GC-MS Method

The Headspace Solid Phase Micro-Extraction-Gas-Chromatography/Mass Spectrometry (HS-SPME-GC-MS) method for furan-derived compounds was modified from Burin et al. [11]—furfural, homofuraneol, furfuryl ethyl ether, ethyl-2-furoate, and 2-furfurylthiol were added to the existing list of analytes. Analyses were carried out on an Agilent Technologies 7890B GC system (Agilent Technologies Inc., Santa Clara, CA, USA), coupled to an Agilent 5977B quadrupole mass spectrometer equipped with a PAL RSI 85 autosampler (Agilent Technologies Inc., Santa Clara, CA, USA). An 85 µm carboxen/polydimethylsiloxane (CAR/PDMS) metal alloy SPME 23-gauge fiber assembly (Sigma Aldrich, St. Louis, MO, USA) was used for sampling. A DB-624UI capillary column was used (30 m × 0.25 mm, 1.4 µm film thickness, Agilent technologies Inc., Santa Clara, CA, USA), with the helium carrier gas (Ultra high purity 5.0) flowing at a rate of 1 mL/min. Vials were stored at 4 °C prior to injection using of a cooling plate accessory (model G4565A). Samples were prepared for injection by agitating (250 rpm) at 40 °C for 5 min before being exposed to the SPME fiber for 55 min at 40 °C with continued agitating (250 rpm), followed by desorption in the inlet at 250 °C for 5 min. The column oven temperature program was: initial temperature 40 °C for 4 min, then raised at 2 °C/min to 160 °C and held for 1 min, and finally ramped to 230 °C at a rate of 5 °C/min, and held at that temperature for 5 min. The total run time for this method was 84 min. The acquisition mode was selective-ion monitoring (SIM), and the interface was kept at 280 °C using electron impact (70 eV) ionization mode. Prior to quantification in SIM mode, a scan was performed (40–300 *m*/*z*) for the identification of target compounds. Analytes and the internal standard were identified according to the ions in Table 4, using OpenLAB CDS Acquisition Agilent Software (Version 2.4.5.9, Agilent Technologies Inc., Santa Clara, CA, USA). The quantifying ions were manually extracted by comparing them to standard peaks, and the ratio of the standard over the internal standard was plotted against the concentration of the compound to fit a linear regression equation.

#### 4.5.3. Analytical Performance of the HS-SPME-GC/MS Method

#### Linearity and Limits of Detection (LOD) and Quantification (LOQ)

The linearity of the HS-SPME-GC/MS method was evaluated using the appropriate concentration range for sparkling wine samples determined by Burin et al. [12]. A low-aroma concentration white wine, which had been dearomatized by rotary evaporation was spiked with the target compounds listed in Table 4, using 6 or 7 concentration levels (excluding 0 µg/L) depending on the peak limit of the instrument, with samples being prepared in duplicate. For all compounds, the standard concentrations were the following: 0 µg/L, 1 µg/L, 5 µg/L, 10 µg/L, 25 µg/L, 50 µg/L, 75 µg/L, and 300 µg/L. The calibration curves were generated by plotting the RF of the analyte (analyte peak area over internal standard peak area) as a function of the analyte concentration. The linearity for all standards was considered satisfactory, with regression coefficients (*R*^2^) greater than 0.99 in all cases. LOD and LOQ’s can be found in Table 4.

#### Repeatability, Accuracy and Specificity

The repeatability of the method was carried out by comparing the relative standard deviations of the standards and samples tested. The coefficients of variation for all standards except for furfurylthiol and 2,3-dihydrobenzofuran were below 10%, which verifies good precision of the method. The accuracy of the method was determined by calculating the percent recoveries for the spiked samples compared to spike blanks. The analyte standards were spiked with 0.1 mL of a 100 µg/L composite standard solution (2 µg/L final analyte concentration) in a de-aromatized real wine sample in duplicate. These spiked samples were then compared to a spike blank dearomatized real wine sample, which had instead been spiked with the 0.1 mL of a 10% ethanol solution, and the relative concentration of the analyte in the spiked sample was compared to that of the spike blank. Recoveries for furfural (106.6%), ethyl-2-furoate (97.9%), and homofuraneol (99.4%) were considered appropriate, and show good accuracy of the method.

### 4.6. Statistical Analyses

Statistical analysis was carried out using XLSTAT (Version 2021.1.1, Addinsoft, Paris, France) for Excel (Version 16.0.14326 for Windows 10, Microsoft Corporation, Redmond, WA, USA). For all data, the Shapiro-Wilk test for normality was performed before any analysis took place to determine which statistical model would be applied. For the total HCA estimation, degree of browning, and LC-UV/DAD data, two-way analysis of variance (ANOVA) and Tukey’s honestly significant difference (HSD) test was used to separate sample means for normally-distributed data, while the Kruskal-Wallis test followed by the Conover-Iman procedure was used for not normally-distributed data, at α = 0.05. For both statistical models, analyte response/spectroscopic reading was the dependent variable, while treatment and temperature were the independent categorical variables. For the HS-SPME-GC/MS and UV-Vis Spectroscopic data analysis, a one-way ANOVA with Tukey’s HSD_0.05_ test was performed on normally-distributed data, while the Kruskal-Wallis test followed by the Conover-Iman procedure was used for not normally-distributed data, at α = 0.05. All graphs and tables were generated using Microsoft Excel (Version 16.0.14326 for Windows 10, Microsoft Corporation, Redmond, WA, USA). The percent difference of compounds stored at 30 °C and 15 °C was calculated as follows: |A − B|/[(A + B)/2] × 100%, where A is the compound concentration determined for the 30 °C sample, and B is the concentration determined for the 15 °C sample. +X% is interpreted as a relatively greater concentration determined at 30 °C compared to 15 °C, while −X% is interpreted as a relatively lower concentration.

## 5. Conclusions

The addition of 5 mg/L of caffeic acid to Chardonnay base wine stored at cellar temperature (15 °C) for 90-days increased the synthesis of furfural by 15% relative to the control. Homofuraneol concentration was also increased by 31% in the caffeic acid-treated wine (20 µg/L) compared to the control (15 µg/L), though wines were above the ODT for this compound. When storage temperature was increased to 30 °C, a smaller increase in furfural (8%) occurred for caffeic acid-treated wine, while no increase was reported for homofuraneol. This shows a direct synergistic relationship between caffeic acid and furfural/homofuraneol formation in wine. The addition of 6 g/L of fructose to the base wine resulted in less significant increases in furan-derivatives than anticipated, a phenomenon that requires further investigation if the synthesis of furan-derived compounds in wines are to be better understood. Additionally, the formation of ethyl-2-furoate was improved in wines kept at 15 °C compared to 30 °C by up to 50% in the control wine, although the concentration of this compound did not exceed 3 µg/L. The storage of base wines/reserve wines for a period of three months or more may be a viable method to encourage the formation of empyreumatic aroma compounds in the final wine. Alternatively, artificially heating base wine for a prolonged period (four months) has been shown to increase furan-derived compounds, and as a consequence could be considered for small batches of base wine in a winery prior to blending to enhance the wine’s complexity.

## Figures and Tables

**Figure 1 molecules-27-07891-f001:**
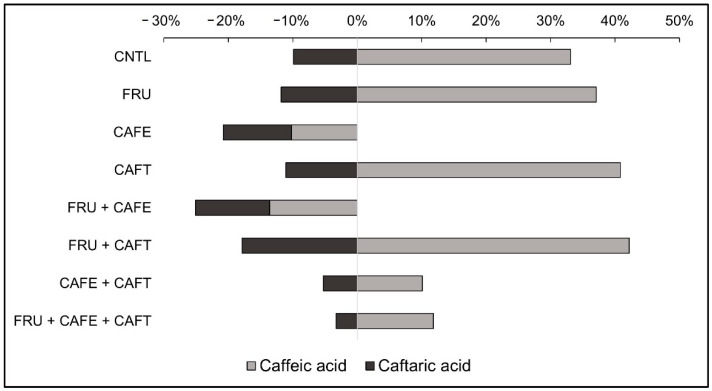
Percent difference in caffeic and caftaric acid concentrations between 30 °C and 15 °C temperature conditions after 90-days of storage. Percentage difference calculated as follows: |A − B|/[(A + B)/2] × 100%, where A is the compound concentration determined for the 30 °C sample, and B is the concentration determined for the 15 °C sample. +X% is interpreted as a relatively greater concentration determined at 30 °C compared to 15 °C, while −X% is interpreted as a relatively lower concentration. Treatment codes: CNTL (no addition), FRU (6 g/L fructose), CAFE (5 mg/L caffeic acid), CAFT (5 mg/L caftaric acid), FRU + CAFE (6 g/L fructose + 5 mg/L caffeic acid), FRU + CAFT (6 g/L fructose + 5 mg/L caftaric acid), CAFE + CAFT (5 mg/L caffeic acid + 5 mg/L caftaric acid), and FRU + CAFE + CAFT (6 g/L fructose + 5 mg/L caffeic acid + 5 mg/L caftaric acid).

**Figure 2 molecules-27-07891-f002:**
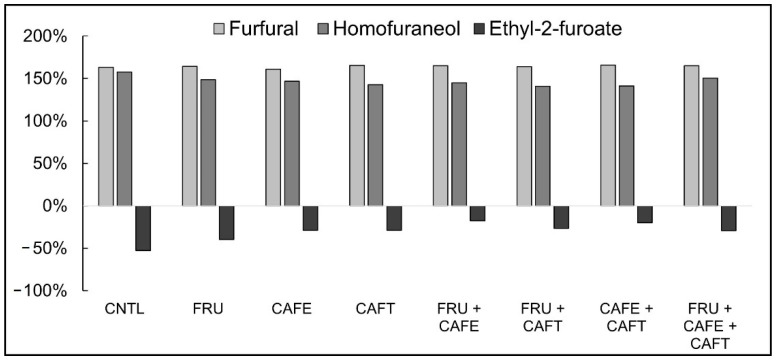
Percent difference between furan-derived compound concentrations stored at 30 °C and 15 °C for 90-days. Percent difference was calculated as follows: |A − B|/[(A + B)/2] × 100%, where A is the compound concentration determined for the 30 °C sample, and B is the concentration determined for the 15 °C sample. +X% is interpreted as a greater concentration determined at 30 °C compared to 15 °C, while −X% is interpreted as a lower concentration. Treatment codes: CNTL (no addition), FRU (6 g/L fructose), CAFE (5 mg/L caffeic acid), CAFT (5 mg/L caftaric acid), FRU + CAFE (6 g/L fructose + 5 mg/L caffeic acid), FRU + CAFT (6 g/L fructose + 5 mg/L caftaric acid), CAFE + CAFT (5 mg/L caffeic acid + 5 mg/L caftaric acid), and FRU + CAFE + CAFT (6 g/L fructose + 5 mg/L caffeic acid + 5 mg/L caftaric acid).

**Table 1 molecules-27-07891-t001:** Caffeic and caftaric acid concentrations determined via Liquid Chromatography-UV Diode Array Detection (LC-UV/DAD) after treatment additions to base wines at the start (Day 0) and end (Day 90) of the temperature-controlled storage period.

Treatment	Caffeic Acid (mg/L)				Caftaric Acid (mg/L)			
	Day 0		Day 90		Day 0		Day 90	
	15 °C	30 °C	15 °C	30 °C	15 °C	30 °C	15 °C	30 °C
CNTL	0.8 ± 0.0 efg	0.8 ± 0.0 g	0.6 ± 0.0 e	0.8 ± 0.0 d	21 ± 0 d	21 ± 0 d	28 ± 0 e	25 ± 0 f
FRU	0.8 ± 0.0 efg	0.8 ± 0.0 fg	0.5 ± 0.0 e	0.8 ± 0.0 d	21 ± 0 d	21 ± 0 d	28 ± 1 e	25 ± 0 f
CAFE	6.0 ± 0.1 bc	5.8 ± 0.2 cd	5.2 ± 0.1 b	4.7 ± 0.1 c	21 ± 0 d	21 ± 1 d	28 ± 0 e	25 ± 0 f
CAFT	0.8 ± 0.0 ef	0.8 ± 0.0 efg	0.6 ± 0.0 e	0.9 ± 0.0 d	28 ± 0 a	26 ± 0 c	35 ± 4 abc	31 ± 0 d
FRU + CAFE	6.0 ± 0.1 c	5.7 ± 0.1 d	5.3 ± 0.1 b	4.6 ± 0.0 c	21 ± 0 d	20 ± 0 d	28 ± 0 e	25 ± 0 f
FRU + CAFT	0.8 ± 0.0 e	0.8 ± 0.0 fg	0.6 ± 0.0 e	0.9 ± 0.0 d	28 ± 1 a	26 ± 0 bc	37 ± 1 a	31 ± 0 d
CAFE + CAFT	5.9 ± 0.1 cd	6.7 ± 0.1 ab	5.1 ± 0.0 b	5.7 ± 0.1 a	26 ± 0 bc	28 ± 0 ab	34 ± 0 ab	33 ± 0 cd
FRU + CAFE + CAFT	6.0 ± 0.0 c	7.0 ± 0.2 a	5.2 ± 0.1 b	5.8 ± 0.1 a	26 ± 0 bc	28 ± 1 a	34 ± 0 abc	33 ± 0 bc

Treatment codes: CNTL (no addition), FRU (6 g/L fructose), CAFE (5 mg/L caffeic acid), CAFT (5 mg/L caftaric acid), FRU + CAFE (6 g/L fructose + 5 mg/L caffeic acid), FRU + CAFT (6 g/L fructose + 5 mg/L caftaric acid), CAFE + CAFT (5 mg/L caffeic acid + 5 mg/L caftaric acid), and FRU + CAFE + CAFT (6 g/L fructose + 5 mg/L caffeic acid + 5 mg/L caftaric acid). Multiple comparison of treatment means at each time point was carried out via the Kruskal-Wallis test for non-parametric data followed by the Conover-Iman procedure. Two-way ANOVAs were performed for parametric data followed by Tukey’s HSD test, using treatment and storage temperature as independent variables. Different letters represent whether means differ as determined by the pairwise comparison of sample means (*n* = 4). ± represents the standard deviation of sample means.

**Table 2 molecules-27-07891-t002:** Furan derivatives determined by Headspace Solid Phase Micro-Extraction-Gas-Chromatography/Mass Spectrometry (HS-SPME-GC/MS) in base wines after 90 days of temperature-controlled storage.

Treatment	Furfural (μg/L)		Ethyl-2-Furoate (μg/L)		Homofuraneol (μg/L)		5-Methyl-Furfural (μg/L)
	15 °C	30 °C	15 °C	30 °C	15 °C	30 °C	30 °C
CNTL	9.1 ± 0.3 b	90 ± 1 b	2.7 ± 0.0	1.5 ± 0.0 d	15.1 ± 0.9 c	128 ± 1	<LOQ
FRU	9.5 ± 0.3 ab	98 ± 0 a	2.8 ± 0.0	1.8 ± 0.1 cd	18.9 ± 0.0 ab	128 ± 4	<LOQ
CAFE	10.5 ± 1.3 a	97 ± 1 a	2.8 ± 0.1	2.1 ± 0.0 bc	19.8 ± 0.3 ab	129 ± 2	4.9 ± 0.0
CAFT	9.2 ± 0.4 b	98 ± 1 a	2.8 ± 0.2	2.1 ± 0.0 bc	21.0 ± 1.7 a	125 ± 5	4.9 ± 0.0
FRU + CAFE	9.5 ± 0.3 ab	99 ± 1 a	2.9 ± 0.0	2.4 ± 0.3 ab	19.8 ± 0.2 ab	125 ± 2	4.9 ± 0.0
FRU + CAFT	9.6 ± 0.3 ab	97 ± 0 a	2.9 ± 0.1	2.2 ± 0.0 ab	20.2 ± 0.1 ab	117 ± 5	5.0 ± 0.1
CAFE + CAFT	9.0 ± 0.2 b	97 ± 1 a	2.9 ± 0.1	2.4 ± 0.1 a	20.1 ± 0.6 ab	117 ± 0	5.0 ± 0.0
FRU + CAFE + CAFT	9.3 ± 0.1 b	97 ± 2 a	2.9 ± 0.0	2.2 ± 0.0 b	18.2 ± 0.4 b	129 ± 3	<LOQ
Significance (between treatments)	*	**	NS	*	**	NS	NS

Treatment codes: CNTL (no addition), FRU (6 g/L fructose), CAFE (5 mg/L caffeic acid), CAFT (5 mg/L caftaric acid), FRU + CAFE (6 g/L fructose + 5 mg/L caffeic acid), FRU + CAFT (6 g/L fructose + 5 mg/L caftaric acid), CAFE + CAFT (5 mg/L caffeic acid + 5 mg/L caftaric acid), and FRU + CAFE + CAFT (6 g/L fructose + 5 mg/L caffeic acid + 5 mg/L caftaric acid). ± represents the standard deviation of sample means (*n* = 4). Different letters represent whether means differ as determined by one-way ANOVA followed by Tukey’s honestly significant difference (HSD) test. Significance: NS = *p* > 0.05, * = *p* < 0.05, ** = *p* < 0.01.

**Table 3 molecules-27-07891-t003:** Base wine chemical composition analyzed prior to treatment additions.

Chemical Parameter	2019 Chardonnay Base Wine
pH	2.89 ± 0.01
Titratable acidity (TA g/L)	10.6 ± 0.1
Fructose (g/L)	2.2 ± 0.1
Glucose (g/L)	0.2 ± 0.0
Malic acid (g/L)	5.3 ± 0.1
Acetic acid (g/L)	0.1 ± 0.0
YAN (mg N/L)	25 ± 1
Alcohol (% *v*/*v*)	10.6 ± 0.1
Free SO_2_ (ppm)	11 ± 0
Total SO_2_ (ppm)	88 ± 5

± represents the standard deviation between sample means (*n* = 2).

**Table 4 molecules-27-07891-t004:** Retention times (min), quantifying ions (*m*/*z*), qualifying ions (*m*/*z*), regression coefficient (*R*^2^), calibration range (µg/L), limits of detection (LODs), and limits of quantification (LOQs) of all furan-derived compounds analyzed.

Compound	Retention Time (min)	Quantifying Ion (*m*/*z*)	Qualifying Ion(s) (*m*/*z*)	Regression Coefficient (*R*^2^)	Calibration Range (µg/L)	LOD ^a^ (µg/L)	LOQ ^b^ (µg/L)
Furfural-d_4_	24.0	100	70, 99	-	-	-	-
Furfural	24.0	96	67, 95	0.9999	2.5–300	0.76	2.5
Ethyl-2-furoate	41.2	95	112, 140	0.9978	1.76–75	0.53	1.76
Homofuraneol	24.0	97	101	0.9995	1.59–300	0.48	1.59
5-methyl furfural	35.2	110	53	0.9988	4.9–300	1.47	4.9
Furfuryl ethyl ether	27.2	81	98, 126	0.9999	1.09–300	0.33	1.09
3-acetyl-2,3-dimethyl furan	43.4	123	138	0.9996	1.00–50	0.30	1.00
2-furyl methyl ketone	30.3	110	95	0.9995	1.03–300	0.31	1.03
2-furfuryl thiol	28.2	81	85	0.9942	4.79–75	1.45	4.79
2,3-dihydro benzofuran	42.5	91	65, 120	0.9999	0.36–50	0.11	0.36

^a^ LODs were calculated by adding the LoB to the value of the lowest concentration analyte replicates’ standard deviation multiplied by 1.645 (LOD = LoB + 1.645 (SD_low conc. sample_); ^b^ LOQs were calculated as the LOD multiplied by 3.3 [32].

## Data Availability

The data presented in this study are available on request from the corresponding author.

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
