# Peer review of "Influence of Caffeic and Caftaric Acid, Fructose, and Storage Temperature on Furan Derivatives in Base Wine"

_molecules, 2022, doi:10.3390/molecules27227891_

Round 1

Reviewer 1 Report

The aim of article "Influence of caffeic and caftaric acid.... "was to determine the extent to which caffeic acid, and its tartaric acid-ester caftaric acid, influence the formation of Maillard-Reaction associated furan-derivative compounds during base wine storage. To assess this, quantification of the furan-derivative compounds was carried out using HS-SPME-GC-MS, in addition to the caffeic and caftaric acids, which were determined by LC-UV/DAD.

            The addition of caffeic acid to Chardonnay base wine stored at 15°C for a 90-day period increased the formation of furfural by 15% relative to the control. Homofuraneol concentration increased by 31% in the caffeic acidtreated . Increasing temperature to 30°C, caused increasing  in furfural of 8% was observed for caffeic acid-treated wine The intentional storage of base wines/reserve wines for a period of three months or more seems  method to encourage the formation of empyreumatic aroma compounds in the final wine. Heating base wine for a longer period causeed increasing furan-derivative compound formation significantly,. It might might be employed for small batches of base wine prior to blending to enhance the wine’s complexity.

The article is preapred correctly. The introduction is correctly written and informative, fully revealing the research aspect.. The Materials and Method , Results are well describe.

Author Response

Thank you to Reviewer 1 for reviewing and commenting on our paper. 

There were no edits to address from Reviewer 1, so we have only addressed the comments from Reviewer 2. 

Regards

Belinda Kemp

Reviewer 2 Report

The submitted manuscript entitled “Influence of caffeic and caftaric acid, fructose, and storage temperature on furan derivatives in base wine” mainly focused on the studies related to assess influences of chemical and temperature on furan products in the wine. Recently, remarkable work has been reported on yeast genetic engineering for the wine improvement. The submitted manuscript has added another perspective for the wine upgrading in all the desirable characters.

The specific comments are as follows :

First three lines in the abstract need be deleted. The proper beginning would be “The aim of this study”. The other text in the abstract has precisely pointed out objectives and results of the undertaken work.

Arrange the keywords in the alphabetical sequence.

It is always advisable to restrict the introduction to the specific topic of the manuscript. Well-documented and widely available data have to be properly abridged in the introduction.

Why there is a need to study effect caffeic and caftaric acid, fructose, and temperature on the formation of  furan-derived compounds? How these studies are necessary? This has to be primarily focused in the introduction.

Table 1. Furan-derived compounds previously identified in wines, is irrelevant in this manuscript, since not exactly related to the reported work in the manuscript.

The last paragraph and preceding paragraphs not judiciously linked to keep a proper flow of the text.  

Section 2.1: Is it necessary to add names of companies providing the chemicals for the work? (Check with the journal’s format).

Section 2.2: Add accession number or any other documentary authentic evidences for Chardonnay grapes documentation.

Wine making is a well standardized practice, write briefly, sub section 2.2 . 

Authors have written extensively on the main experimental protocols, demonstrating the illustrious work carried out by them. However, it is recommended to emphasis on the innovative and novel approaches adopted in the present work. Routinely practiced techniques  be shortened, wherever possible.  

Table 2 titled “Base wine oenological parameters analyzed pre-treatment addition” suitably presented.

A comparative data of Caffeic and Caftaric acid, describing interaction between temperature and added acids have been presented with extensive experimental results (Table 4).

The authors have further extended their studies on Furan-derived compounds, Furfural, Ethyl-2-furoate, Homofuraneol and 5-Methylfurfural. All the studies have been based on the substantial and elaborate experimental data (Table 5, Fig 2). Treatments are logically selected for deriving the convincing conclusions.

Although discussion is fittingly written, it is recommended to reduce the text wherever possible. An abridged and precise discussion would be a better option.

I appreciate that authors have also included the limitations of their studies.

This is a technology  based manuscript and it would be prudent to incorporate the cost calculations of the entire process.

How these studies will help to modify the currently practiced wine making processes?

Some of the references are old, keep only most relevant citations or refer to recently published reviews.

Author Response

Thank you to Reviewer 2 for their time reviewing the manuscript.

Please find attached the response to comments from Reviewer 2.

Regards

Belinda Kemp
